# Evidence for auto-catalytic mineral dissolution from surface-specific vibrational spectroscopy

Jan Schaefer[1], Ellen H.G. Backus[1] & Mischa Bonn[1]

The dissolution of minerals in water is typically studied on macroscopic length- and time-scales, by detecting dissolution products in bulk solution and deducing reaction rates from model assumptions. Here, we report a direct, real-time measurement of silica dissolution, by monitoring how dissolution changes the first few interfacial layers of water in contact with silica, using surface-specific spectroscopy. We obtain direct information on the dissolution kinetics of this geochemically relevant mineral. The interfacial concentration of dissolution products saturates at the level of the solubility limit of silica (~millimolar) on the surprisingly short timescale of tens of hours. The observed kinetics reveal that the dissolution rate increases substantially with progressing dissolution, suggesting that dissolution is an auto-catalytic process.

[1] Max Planck Institute for Polymer Research, Ackermannweg 10, 55128 Mainz, Germany. Correspondence and requests for materials should be addressed to M.B. (email: bonn@mpip-mainz.mpg.de)

nsights into the physicochemical properties of mineral/water interfaces at the molecular level are essential for understanding many of the geological and electrochemical processes on macroscopic scales[1]. For geological questions, it is important to look at these systems under non-equilibrium conditions, as in nature, water is usually not at rest. Typical examples are sea and freshwater dissolving and carrying minerals, which then affects the geology and biology of the environment around it. In this work, we study the dissolution of silica, one of the most abundant minerals on earth, into flowing and resting water.

With bulk experiments[2–5], it has been demonstrated that silica may dissolve up to millimolar concentration of silica species through hydrolysis reactions with interfacial water, over very long (10s–100s of hours) timescales. Very little is known about the dissolution dynamics and how the kinetics affect the structure and composition of the silica/water interface including its physical properties such as the surface charge density, the Debye length, and corresponding surface potential. While silica/water has been studied extensively under static conditions by employing non-linear spectroscopy techniques[5–19], potentiometric titration[3,20], Atomic Force Microscopy[21], and X-ray Photoelectron Spectroscopy[22,23], non-equilibrium insights are scarce[24,25].

Here, we employ vibrational sum frequency generation (V-SFG) spectroscopy, a sub-μm surface-sensitive and chemically selective technique. We use the vibrational response associated with the O–H stretching mode of H-bonded water at ~3200 cm$^{-1}$ to monitor the dissolution kinetics at the silica/water interface. The V-SFG water signal reflects the broken symmetry at the interface: specifically for water near charged interfaces, the surface charge serves to align the water molecules, thereby breaking the symmetry and making the water molecules SFG active. The $z$-dependent surface potential itself also breaks the symmetry, giving rise to a potential signal from the near-surface region, corresponding to the Debye length, with $z$ being the coordinate along the surface normal. As such, the V-SFG interfacial water signal reflects both the surface charge and the extent to which that surface charge is screened by counterions at the surface. We find that equilibration of the silica surface in contact with bulk water occurs on a timescale of several tens of hours, resulting in a steady-state interfacial concentration of ionic dissolution products corresponding to an ionic strength of ~1–10 mM. The dissolution process appears auto-catalytic.

## Results

**Steady-state SFG spectra.** Dissolution of silica changes the ionic strength of interfacial water since some of its dissolution products are known to be charged. Monosilicic acid, for example, may mainly exist as an over-protonated (i.e., positively charged) species at neutral pH[5]. Therefore, dissolution may cause changes in the surface charge of silica as well as the screening of that charge by nearby ions, i.e., the Debye length. The SFG O–H stretch signal is known to be a sensitive reporter of both the surface charge and the Debye length, as explained in detail in ref. [26] and illustrated in Fig. 1. In ref. [26], it is shown that the SFG intensity variation with ionic strength of NaCl can be accounted for by invoking only screening and assuming $\chi^2$ as well as the surface charge, primarily determined by the local pH[27], to be constant[26]. Figure 1 reveals a maximum in the SFG signal as a function of ionic strength (NaCl), around 1 mM. This maximum originates from an interplay between screening effects lowering the signal at higher concentrations (regions ii–iv) and interference effects, lowering the signal at lower concentrations (region i)[26]. There are also subtle differences in the spectral shape of the response, discussed in the Supporting Information and in ref. [26], not relevant to the discussion here.

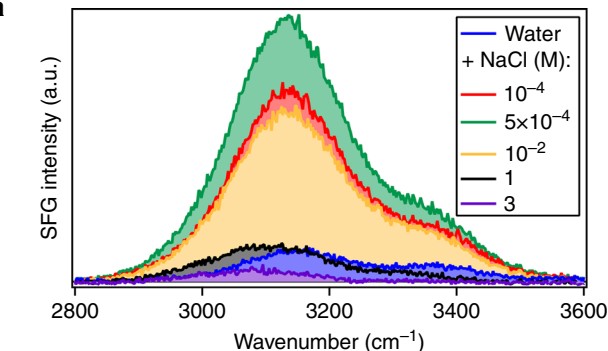

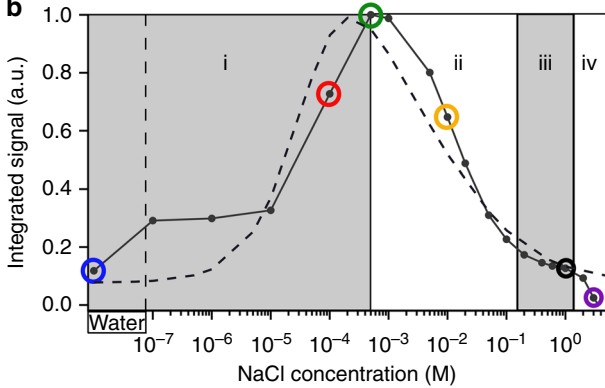

**Fig. 1** SFG spectra. **a** O–H-stretch SFG signal for various NaCl solutions (ref. [26]). **b** SFG signal intensities, integrated from 2800 to 3600 cm$^{-1}$ and normalized to the curve maximum. The spectra corresponding to the marked data points are presented in the top panel, with corresponding colors. All spectra are measured under resting conditions, directly after flowing for 5 min. Dashed line: the predicted intensity variation by invoking modulation by charge screening and interference, at constant $\chi^2$ and constant surface charge density, as detailed in ref. [26]

**Dissolution kinetics probed with SFG.** Here, we use the known correlation between the near-surface charge distribution and the SFG signal for studying the dissolution of silica at the silica/water interface. For doing so, we monitor the O–H stretch V-SFG signal as a function of time after bringing the silica surface in contact with water. By applying flow, i.e., replacing the water in the interfacial region by bulk solution, the interfacial ionic strength becomes dominated by the bulk electrolyte concentration. Through comparison of pure water with various NaCl solutions under resting and flowing conditions, we obtain new insights into the dissolution process at the silica/water interface by quantifying the time-dependent dissolution-generated interfacial ionic strength. For quantification, we need to find the concentration above which we do not observe any signal changes upon flow, which means that the ionic strength of dissolved silica matches or is overwhelmed by the bulk NaCl concentration.

The time-dependent V-SFG responses for pure water, $10^{-4}$, $5\times10^{-4}$, $10^{-2}$, and 1 M NaCl aqueous solution are presented in Fig. 2. Similar to Fig. 1, the V-SFG signals are analyzed by signal area. Overall, we observe that under flow-off conditions, the signal traces for all solutions up to a NaCl concentration of 10 mM converge to the same SFG intensity, which is close to the intensity observed for the 10 mM solution under flow-perturbed conditions. Without invoking any model, this directly indicates that dissolution of silica changes the $z$-dependent electrostatic potential of the bare silica/water interface to a very similar degree as the addition of approximately 10 mM of NaCl, implying an interfacial concentration of dissolution-released ions of that same

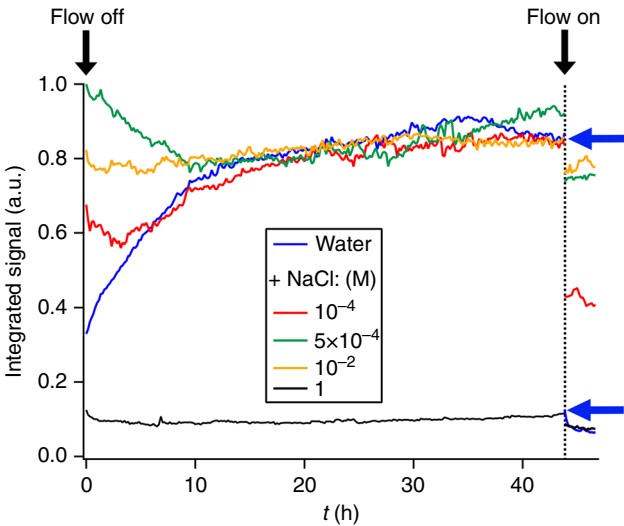

**Fig. 2** Kinetics. Kinetics of the O–H-stretch SFG signal from the silica/water interface at different sodium chloride bulk concentrations under resting ("Flow off") and flowing ("Flow on") conditions

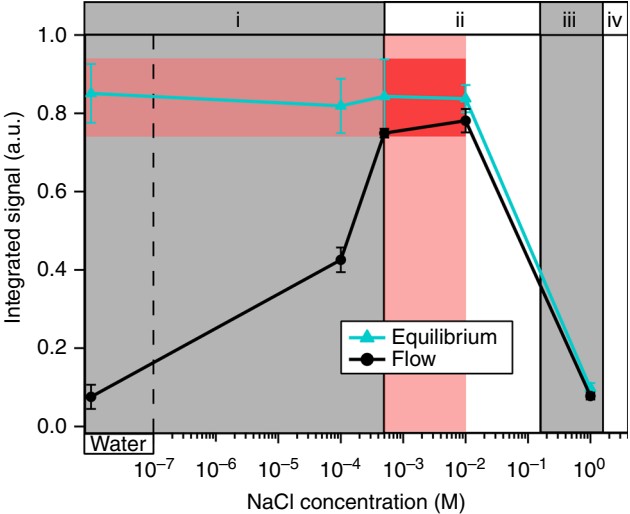

**Fig. 3** Integrated SFG signal. Integrated O–H-stretch SFG signal of the silica/water interface at equilibrium ($t > 12\,h$) and under flow conditions with different bulk sodium chloride concentrations. The error bars represent the $2\sigma$ variation from the corresponding average value

order. A spectral analysis of the signals supports the conclusion that the dissolution products in this concentration range screen the surface charge. The spectral shift of the O–H band corresponds quantitatively to that observed previously between pure water and a salt solution (NaCl) in the millimolar concentration range (detailed in Supplementary Note 1)[26]. As mentioned above, the associated SFG intensity variation of those NaCl solutions can be accounted for by just invoking a concentration-dependent Debye screening length, keeping $\chi^2$ and the surface charge density constant (Fig. 1, dashed line)[26]. This suggests that the intensity change, also along dissolution, primarily results from variation in charge screening. However, we cannot exclude an additional contribution from changes in the surface charge, as indicated in a previous work using a different, more surface-specific, experimental geometry[25].

For silica in contact with pure water, the O–H-stretch SFG signal rises to the equilibrated level within a timescale of ~1 day (blue trace in Fig. 2). Upon turning on the flow at $t = 45\,h$, i.e., by exchanging the interfacial solution with pure water, a rapid (<10 min) intensity drop of about one order of magnitude is observed (blue arrows in Fig. 2). Under resting conditions, dissolving silica generates substantial interfacial ionic strength as reflected by the intensity changes, equivalent to going from the low-concentration (left) end of the trace in Fig. 1 to the peak of the signal at ~1 mM.

For $10^{-4}$ and $5 \times 10^{-4}$ M NaCl solutions, the signals initially decrease within the first 5 and 10 h, respectively. Afterward, both traces follow the kinetics as observed for pure water. Hence, at later times, the ionic strength of the interfacial region is dominated by dissolved silica species whereas the initial decrease can only be ascribed to a process involving NaCl since it does not appear for pure water. One scenario is that at early times ($t < 10\,h$), dissolving silicic acid molecules replace $Na^+$ and $Cl^-$ ions from the interface. In line with this hypothesis is the previous observation that ions with equal valence but larger hydrated radii (here: silicic acid vs. $Na^+/Cl^-$ ions) screen surface charges less efficiently[26]. While less screening usually corresponds to a higher SFG signal, solutions with sub-mM ionic strength show opposite behavior as demonstrated in Fig. 1. In this concentration regime, a decrease of the SFG response reflects an increase of the penetration depth of the surface electric field as a result of reduced charge screening[26,28]. Upon flow, i.e., mixing the

interfacial region with a bulk electrolyte solution, the SFG signals drop according to the bulk ionic strength of the corresponding solution, similar to what is observed for pure water.

For $10^{-2}$ M and 1 M NaCl, the signal strengths are very different, but in both cases, no substantial intensity change occurs within 45 h of equilibration which means that the screening length, determined by the interfacial ionic strength, in both cases is dominated by the NaCl content in the bulk solution. However, under flow conditions, a small intensity drop can still be observed for both solutions. One explanation could be that application of flow does not only mix interfacial with bulk solution, but may also slightly alter the effective surface charge density of silica[25].

## Discussion

As summarized in Fig. 3, the differences between equilibrium and the flow-perturbed steady states increase with decreasing electrolyte concentration. This observation is consistent with the notion that the intensity rise during equilibration can be ascribed to a local increase of ionic concentration close to the silica surface. Under flow conditions, the local ionic strength at the surface is decreased if the equilibrium concentration of silicic acid exceeds the bulk ionic concentration. For increasing bulk ionic concentration, the removal of dissolved silica is compensated by adsorption of ions provided by the bulk solution, effectively decreasing the difference in signal intensity along equilibration and under flow conditions.

It is apparent from these measurements that the V-SFG intensity of the nominally pure water sample under equilibrated conditions is equal to the intensities observed for the solutions with bulk NaCl concentrations between $5 \times 10^{-4}$ and $10^{-2}$ M, which is very similar to the solubility limit of silicic acid[2,3,5]. Apparently, the interfacial ionic strength, generated by silica dissolution, reaches a steady state within this concentration range. In contrast to the bulk solutions of high-surface-area silica, for which it is known to take roughly 2 weeks to achieve saturation with silicic acid[5], our results show that the interfacial region is already saturated after 1 day. This observation suggests that diffusion limits dissolution of silica in contact with resting water.

A quantitative picture of the interfacial kinetics can be obtained by invoking a simple one-dimensional (1D) dissolution rate

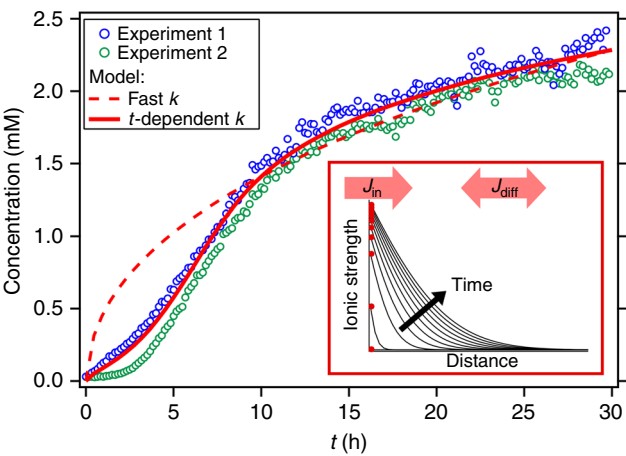

**Fig. 4** Time trace. The time trace of the O–H-stretch SFG signal from the silica/water interface for pure water converted into an interfacial ionic strength (circles) by using the conversion function detailed in Supplementary Note 2. For comparison, the interfacial ionic strength as predicted by a simple 1D dissolution/diffusion model (red lines). Inset: a schematic plot of the concentration gradient perpendicular to the surface plane resulting from the model, plotted at equally distant times. The red dots represent the evolution of the interfacial ionic strength (average value of the first 1 μm)

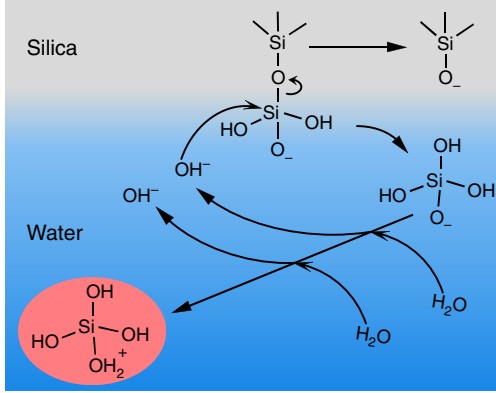

**Fig. 5** Proposed dissolution mechanism of silica in water considering hydrolysis with hydroxide ions. With each reaction cycle, protonation of the dissolved silicic acid molecule releases two hydroxide ions close to the surface which are mostly consumed by the hydrolysis reaction, giving rise to an auto-catalytic process

model that considers only one type of silica species, being either bound to the solid or dissolved in water and getting displaced by diffusion with the corresponding overall forward and backward rates $k_f$, $k_b$ and the diffusion coefficient ($D$). Equations 1, 2 provide the expressions for the corresponding interfacial and diffusive flux. The interfacial flux ($J_{in}$) depends on $k_f$, $k_b$, and the interfacial concentration of dissolved species ($c$). The diffusive flux ($J_{diff}$) depends on $D$ and the concentration gradient orthogonal to the surface plane.

$$J_{in} = k_f - k_b c. \qquad (1)$$

$$J_{diff} = \frac{\partial c}{\partial t} = D \frac{\partial^2 c}{\partial z^2}. \qquad (2)$$

The rate model results plotted in Fig. 4 present the temporal change of the ionic strength within the first micrometer from the interface and the spatial gradient orthogonal to the surface plane (see inset). The main plot compares the temporal change of the interfacial concentration as resulting from the model (red lines) with two experimental time traces for nominally pure water (blue and green). The experimental concentration is inferred from the integrated O–H stretch V-SFG signal, converted by a function that resembles the low concentration intensity-curve shape (detailed in Supplementary Note 2).

By using rates of the order of magnitude as reported in the literature, $k_f$ = 2.25×10$^{-12}$ mol m$^{-2}$ s$^{-1}$ (refs. [5,29–31]), $k_b$ = 1.90×10$^{-10}$ s$^{-1}$ (ref. [4]), $D$ = 8.50×10$^{-10}$ m$^2$ s$^{-1}$ (ref. [32]), the model does not result in an interfacial ionic strength on the millimolar range. This demonstrates that these dissolution rates, deduced from bulk experiments, only yield an accurate description of the dissolution process under continuous mixing, while it poorly describes the actual reaction, that is, under resting conditions. For yielding a millimolar concentration on the timescale of a few hours (red dashed line), the rates have to be adjusted substantially, either by accelerating the dissolution ($k_f$ = 2.40×10$^{-10}$ mol m$^{-2}$ s$^{-1}$, $k_b$ = 4.72×10$^{-8}$ s$^{-1}$) or by slowing down the diffusion ($D$ = 9.50×10$^{-14}$ m$^2$ s$^{-1}$). However,

neither adjustment yields the correct shape of the experimental traces that show a slow response at early times ($t < 5$ h), followed by an accelerated concentration increase ($5$ h $< t < 10$ h) and saturation at late times ($t > 10$ h). The observation of a lag time for the interfacial concentration to start changing unambiguously shows that there is a time-dependent contribution to the dissolution process, which is missed in bulk experiments[5,29–31]. Potentially, this may arise from a time-dependent diffusion coefficient as detailed in Supplementary Note 3. A physically more compelling interpretation is a time-dependence of the reaction rate, discussed in the following.

The S-type experimental response can be understood by considering a reaction pathway that allows for a self-accelerating dissolution process of silica, namely hydrolysis with hydroxide ions. Hydroxide ions are released during the reaction, and the dissolution rate has previously been reported to increase at elevated pH[5].

$$\equiv Si-O-Si(OH)_3 + OH^- \rightleftharpoons \equiv Si-O-Si(OH)_2 O^- + H_2O \qquad (3)$$

$$\equiv Si-O-Si(OH)_2 O^- + OH^- \rightleftharpoons \equiv Si-O^- + Si(OH)_3 O^- \qquad (4)$$

At neutral to slightly acidic pH, dissolved silicic acid mainly exists as an over-protonated species which returns two hydroxides[5].

$$Si(OH)_3 O^- + 2 H_2O \rightleftharpoons Si(OH)_4 H^+ + 2 OH^- \qquad (5)$$

Given that (de-)protonation occurs on a much faster timescale than dissolution, the hydrolysis is the rate-limiting step and determines the overall rate. At the same time, protonation of each dissolved silicic acid molecule returns two hydroxides that catalyze dissolution of the subsequent layer. As summarized schematically in Fig. 5, the dissolution of silicic acid thus is an auto-catalyzed process that accelerates as the dissolution proceeds. We performed a control experiment with a pH 9 aqueous solution of NaOH (Supplementary Note 4). In line with dissolution being an auto-catalytic process, with hydroxyl ions being the key reactant, we observe that the kinetics speed up by over an order of magnitude: from a tens of hours timescale to a 1–2-h timescale under basic conditions. This observation supports the notion of auto-catalytic, hydroxide-driven dissolution of silica.

In this scenario, the overall rate largely depends on where, relative to the surface, the dissolved silicic acid molecules protonate, determining the local concentration of hydroxides, which

induce deprotonation and dissolution of the subsequent surface layer. With increasing number of hydroxides present at the surface, the rate will increase. Invoking a linear increase of $k_f$ with time (from $k_{f,0h} = 1.01 \times 10^{-12}$ to $k_{f,8.5h} = 3.69 \times 10^{-10}$, detailed in Supplementary Note 5), the model yields the S-type curve shape (solid red line in Fig. 4) as observed experimentally.

In conclusion, we have experimentally demonstrated that silica equilibrates roughly at a millimolar of ionic strength at the interface with pure water. This interfacial equilibration takes tens of hours, not weeks as concluded from bulk experiments with high surface area silica, suggesting that the overall dissolution process of silica is limited by diffusion. This is supported by the fact that mixing, as often done in those bulk experiments and accomplished here by applying flow, rapidly ($t < 10$ min) yields a steady state in which dissolution products are instantaneously removed from the very first interfacial layers probed in our experiments.

By comparison with a 1D reaction–diffusion model, we show that dissolution rates and a diffusion coefficient reported for bulk dissolution of silica do not yield sufficiently high equilibrium interfacial ionic strength as observed in our experiments, at least within the timescale of hours. The experimentally observed S-shaped kinetic curve directly shows that the dissolution rate of silica increases with time, suggesting an auto-catalyzed mechanism.

## Methods

**SFG spectroscopy**. The SFG spectra presented in this work were recorded with an experimental setting that is explained in detail in ref. [26] and based on a Ti:sapphire regenerative amplifier (Solstice® Ace™, Spectra Physics), generating 800 nm pulses with ~40 fs duration and 1 kHz repetition rate. A commercial optical parametric amplifier (TOPAS Prime, Spectra Physics) and a non-collinear difference frequency generation (NDFG) scheme was used to generate broadband (~450 cm$^{-1}$) infrared pulses. The SFG signal was spectrally resolved and detected using a spectrograph (Acton Spectro Pro® SP-2300, Princeton Instruments) together with an emCCD camera (ProEM 1600, Roper Scientific). All spectra were measured with ssp polarization combination (s-polarized SFG, s-polarized visible, and p-polarized IR). To check laser stability during the experiment, part of the IR beam was split off to monitor its power. A typical time trace is plotted in the bottom panel of Supplementary Note 6, showing a variation of less than 5%.

**Sample preparation**. Sodium chloride, purchased from Sigma Aldrich (Z99.5%, CAS 7647-14-5), was baked at 650 °C for 6 h to remove organic residues and dissolved in demineralized H$_2$O, filtered with a Millipore unit (resistivity = 18 MΩ cm). Before each experiment series, the (parallel faced) fused silica window (Korth Kristalle GmbH Infrasil 302, s/d: 60/40) was treated by UV–Ozone cleaning for 30 min. To ensure CO$_2$-equilibration, all sample solutions were measured at least 1 h after preparation. Before each measurement, the cell (introduced in ref. [25]) was flushed with the sample solution ($V_{total} = 250$ mL) for 5 min (500 mL min$^{-1}$). The same flow rate was used for carrying out the flow-perturbed experiments.

**Model**. The 1D reaction–diffusion model was implemented by employing the built-in MATLAB (R2017a) scheme pdepe.

**Data availability**. The authors declare that the data supporting the findings of this work are available within the paper and its Supplementary Information files or from the corresponding author on reasonable request.

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

## Acknowledgements

We thank Dan Lis for initiating this project as well as Sapun Parekh, Johannes Hunger, Kaloian Koynov, and Stoyan Yordanov for fruitful discussions. Additionally, we thank Johannes Hunger for a careful reading of the manuscript. This work was funded by an ERC Starting Grant (Grant No. 336679).

## Author contributions

J.S., E.H.G.B. and M.B. designed the research project. J.S. performed the experiments and the simulation. All authors discussed the results and wrote the manuscript.

## Additional information

**Competing interests:** The authors declare no competing interests.



