## [Peer Review File · Nature Communications]

Reviewers' comments:

Reviewer #1 (Remarks to the Author):

This paper studies the silica/water interface using vibrational sum frequency generation in the presence of low and high salt concentrations under equilibrium and non-equilibrium conditions to identify the dissolution dynamics of silica in water on the microscopic scale. The authors draw back to previous work on screening and interference effects (Figure 1) as a sort of calibration curve to estimate the interfacial electrolyte concentration in their experiment. Among key points made are that silica dissolution is an auto-catalytic process driven by the formation of silicic acid and hydroxide which re-enters the catalytic cycle, and that interfacial equilibrium of dissolved species occurs on the order of tens of hours instead of several weeks. The paper presents intriguing evidence of an auto-catalytic dissolution mechanism as well as far faster equilibration of the interfacial region in comparison to what is observed in bulk dissolution experiments.

This paper is important and is of the high quality and novelty expected for Nature Communications. It should be accepted for publication after minor revisions and the following comments should be addressed.

Main Comments:

- 1) In the introduction (line 40) the authors mention that the mode at $\sim 3200\text{ cm}^{-1}$ is monitored, but in the rest of the paper it is stated that integrated values are used. Given that the authors mention that there are subtle differences in the salt-dependent spectral shape of the OH stretching region, could the authors specify the range over which the spectra were integrated?
- 2) The author's finding that silica dissolution occurs on the order of tens of hours is very interesting. Although the authors comment that diffusion is likely the reason for the difference in timescales for the interface to reach equilibrium versus the bulk, the authors should add some discussion on whether this is expected based on current understanding of mass transport at these interfaces.
- 3) Could the authors comment on why the integrated SFG of silica/water in the OH stretching region for pure water is lower (or similar to) than that of 100 mM NaCl (Figure 1)? This is surprising, as usually the presence of such high concentrations of NaCl decreases the overall signal as has been shown in ref 8 among others.
- 4) The auto-catalytic mechanism requires a net increase in catalyst (hydroxide) as the reaction progresses, but this is not implicitly shown as Figure 5 illustrates two hydroxides being used to generate one dissolved deprotonated silicic acid, which results in the generation of two hydroxides after protonation of the solution species (two used \Rightarrow two generated). I think the authors need to show how the remaining surface siloxide formed after release of the silicic acid can react with water to form some kind of species like that formed in equation 3. This bypasses the need for hydroxide in that step which means that there would be a net increase in hydroxide once dissolution of that group takes place (one hydroxide used, two generated by the released silicic acid). In other words, the scheme shows hydroxide being catalytic, but another step needs to be added to show a net increase in hydroxide which backs up the observations of auto-catalysis.

Small things:

Page 2, line 37: should be "insights are scarce".

The cited Reference 17 is only the correction to the paper and not the original paper. Perhaps both should be cited?

Reviewer #2 (Remarks to the Author):

The manuscript by Schaefer et al reports an investigation on dissociation of silica surface in contact with water by surface-specific vibrational spectroscopy. The authors evaluated the interfacial ionic concentration arisen from dissolution products and the associated reaction kinetics through examining time-dependent SFG intensity and the effect of liquid flow. While the experimental results appear to have far-reaching implications for interfacial chemistry, I have concerns about analysis and interpretation, as describing in details below. Since it is crucial for validity of the main conclusions of the manuscript, I think that the manuscript is not suitable for publication with the current form.

1. The experiments in the manuscript are mainly explained by assuming an insignificant change in the surface charge density of the silica/water interface, but this assumption is not justified nor discussed in this manuscript. (nor in Ref. 24, where Fig. 1 of this manuscript comes from). In particular, the authors make this assumption in line 60 and 61 by considering the local (surface) pH as the primary factor determining the surface charge density. This argument is not sufficient to support the authors' assumption. It is because the surface pH is associated with the interfacial potential (Ref. 25), and the interfacial potential changes upon different ionic concentrations in the presented experiments. It is, therefore, questionable whether the variations of local pH and the consequent surface charge density can be assumed insignificant in interpreting the experiments. Note that possible changes of the surface charge density could affect the interfacial ionic concentration deduced in Fig. 4 (deduced under assumption of constant surface charge density) and could, therefore, affect validity of the authors' subsequent discussions about kinetics and pathway of the reaction (Fig. 5).

2. The auto-catalytic process proposed in the manuscript generates a negatively charged Si-O- group and an over-protonated monosilicic acid during one cycle of the reaction (Eq. 3-5). The authors used screening effect of the over-protonated monosilicic acid to explain the time-dependent SFG measurement in Fig. 2. The negatively charged byproducts, i.e., negatively charged Si-O- groups, with the equal amount are then expected to appear at the interface and thus contribute to the surface charge density. It is unclear whether or not the surface Si-O- groups generated from the auto-catalytic process induce minor changes of the surface charge density, as assumed in the manuscript. This issue needs to be addressed to ensure self-consistency of the data interpretation.

3. The authors have to address (in)consistency of results presented in the manuscript with respect to these in their previous reports. Some important information are missing. For instance, a figure from Ref. 24 is used as Fig. 1 in the presented manuscript to discuss the salt-concentration effect. It is unclear whether Fig. 1 was measured under resting or flowing condition, and whether Fig. 1 and Fig. 2 are comparable with each other. In addition, in a previous paper published by the same group (ref 23), the authors observed ~50% difference in the SFG intensity under resting and flowing conditions in similar experimental conditions (silica/water interface, 10mM NaCl, pH=6.5, flow OFF and ON). However, this effect is obviously weaker in the submitted manuscript. (Fig. 2 in yellow, flow OFF and ON at 10 mM, and Fig. 4 at 10mM, the difference is only ~10%). It is also noticed that the data interpretations are different in the authors' different publications. In Ref. 23, the difference of the SFG intensity observed upon flowing of liquids was explained by the change in the surface charge density, and the change of Debye length was considered to be minor. These interpretations are obviously against these in the presented manuscript, without any comment given.

Reviewer #3 (Remarks to the Author):

This paper presents some very interesting experiments in which vibrational sum-frequency

generation is used to probe the silica/water interface over a time scale of many hours as the silica is slowly dissolved. The authors present convincing kinetic data showing that dissolution is considerably faster than has been concluded in other studies, which have focused on bulk rather than interfacial observables. This is a clever and fascinating study that will undoubtedly receive a lot of well-deserved attention.

The part of the work that I am less convinced about is the argument for autocatalysis. The authors observe that the rate of dissolution increases after an induction period, which they convincingly argue cannot be described by the simplest reasonable kinetic model. They propose a plausible mechanism to account for this observation. However, if autocatalysis is going to be a primary focus of the paper (and it is in the title, after all), then some control experiments are called for. First, as they posit that base-catalyzed hydrolysis accounts for their observations, it would seem that if they were to begin their experiments under basic conditions that there would be no induction period. Second, I am not sure what the plasma treatment does to the surface of the silica, but one possibility is that it tends to remove all of the silanol functionality (although this may be restored by contact with water). At any rate, the induction period could certainly have something to do with the surface structure of the silica following plasma treatment, and it is not at all clear to me that the rate-limiting step should be the final one in dissolution. Each Si atom in the bulk is bonded to four O atoms, and I would (admittedly, possibly naively) think that the breaking of the first siloxane bridge for a particular Si atom would be rate limiting. I would suggest two control experiments to test the proposed mechanism further. The first of these the authors may already have performed, which is to determine what happens if they allow the dissolution to reach its maximum rate and then flow pure water into the system. Are the kinetics then reproducible, including the induction period? The second control would be to use a high-temperature vacuum oven instead of plasma treatment to prepare the surface. The vacuum oven will also remove organic contaminants, but will also tend to remove silanol groups so that all of the surface oxygen atoms are involved in siloxane bridges. Looking at the effects of this alternate treatment should reveal if the induction period is related to surface structure.

On a minor additional note, the authors should do another careful proofreading of the paper, as I did come across a number of typographical errors.

Overall, this is a really exciting piece of work and I am strongly in favor of its publication in Nature Communications. I know that no one likes to get a review suggesting additional experiments, but a few more tests could really make the proposed picture of autocatalysis substantially more "bullet-proof," which would enhance the already strong impact of this work considerably.

Response to reviewer 1

We thank the reviewer for carefully reading the manuscript and the constructive, encouraging comments. We have changed the manuscript according to the reviewer's comments.

Below, we give a point-by-point response to the comments of the reviewer, and the action taken. We have reproduced the reviewer's comments in italic font below.

This paper studies the silica/water interface using vibrational sum frequency generation in the presence of low and high salt concentrations under equilibrium and non-equilibrium conditions to identify the dissolution dynamics of silica in water on the microscopic scale. The authors draw back to previous work on screening and interference effects (Figure 1) as a sort of calibration curve to estimate the interfacial electrolyte concentration in their experiment. Among key points made are that silica dissolution is an auto-catalytic process driven by the formation of silicic acid and hydroxide which re-enters the catalytic cycle, and that interfacial equilibrium of dissolved species occurs on the order of tens of hours instead of several weeks. The paper presents intriguing evidence of an auto-catalytic dissolution mechanism as well as far faster equilibration of the interfacial region in comparison to what is observed in bulk dissolution experiments.

This paper is important and is of the high quality and novelty expected for Nature Communications. It should be accepted for publication after minor revisions and the following comments should be addressed.

We appreciate the reviewer's positive assessment of our manuscript.

1) In the introduction (line 40) the authors mention that the mode at $\sim 3200\text{ cm}^{-1}$ is monitored, but in the rest of the paper it is stated that integrated values are used. Given that the authors mention that there are subtle differences in the salt-dependent spectral shape of the OH stretching region, could the authors specify the range over which the spectra were integrated?

In accordance with the reviewer's comment, we changed the figure caption where we now specify the integration window in line 69.

Action: in the caption of figure 1, we have included the comment: "Bottom: SFG signal intensities, integrated from 2800 to 3600 cm^{-1} and normalized to the curve maximum."

2) The author's finding that silica dissolution occurs on the order of tens of hours is very interesting. Although the authors comment that diffusion is likely the reason for the difference in timescales for the interface to reach equilibrium versus the bulk, the authors should add some discussion on whether this is expected based on the current understanding of mass transport at these interfaces.

The reviewer raises an excellent point. In agreement with this suggestion, we have added a discussion to the manuscript (line 180-183 and 191). Indeed, based on the current understanding of mass transport, there are substantial differences in timescales expected for the bulk vs the interface.

However, the bulk-deduced rates stem from experiments where the solution is continuously mixed. As observed in our work, the reaction rate increases in the presence of dissolution products in solution near the surface. Therefore, a bulk experiment (under mixing conditions) inherently probes predominantly the initial (slow) rate. In our interfacial experiments, the solution is not stirred which means that: (i) the dissolution process is limited by mass transport (diffusion); and (ii) the dependence of the rate on the presence of dissolution products can be resolved.

Action: on lines 180 – 183, we have included the comment: “This demonstrates that these dissolution rates, deduced from bulk experiments, only yield an accurate description of the dissolution process under continuous mixing, while it poorly describes the actual reaction, that is, under resting conditions.”, And on line 191: “...shows that there is a time-dependent contribution to the dissolution process, which is missed in bulk experiments.”^{5,29–31,,}

3) Could the authors comment on why the integrated SFG of silica/water in the OH stretching region for pure water is lower (or similar to) than that of 100 mM NaCl (Figure 1)? This is surprising, as usually the presence of such high concentrations of NaCl decreases the overall signal as has been shown in ref 8 among others.

Indeed, this finding is surprising as usually the effect of salt on the SFG response is interpreted differently. A brief explanation had already been provided in the original manuscript on lines 62-66 (line numbers refer to the revised manuscript); a detailed discussion can be found in reference 26, that we now cite in line 65 in agreement with the reviewer’s comment. A brief summary of the phenomenon: In line with the common picture is the concentration regime B-D in Figure 1. But in regime A, $[\text{NaCl}] < 5 \cdot 10^{-4} \text{ M}$, the Debye length exceeds the coherence length of the V-SFG process ($\sim 40 \text{ nm}$) which gives rise to destructive interference and cancellation for the field-induced part of the SFG signal. In other words: SFG is generated in the near-surface region (that region being determined by the Debye length). SFG generated in different spatial positions in the region will interfere, as these differ slightly in phase. For sufficiently low salt concentration, the Debye length becomes large enough, that these signals can cancel out, collectively. This explains the decrease in SFG signal for concentration approaching zero. We have added a model calculation to the data in figure 1, according to reference 26.

Action: we have added, in addition to citing reference 26 in line 66, added in lines 59 – 61, the text: “In ref ²⁶, it is shown that the SFG intensity variation with ionic strength of NaCl can be accounted for by invoking only screening and assuming the surface charge, primarily determined by the local pH,²⁷ to be constant.²⁶”. Moreover, we have added the dashed line to the bottom panel of figure 1, and to the figure caption of figure 1: “Dashed line: The predicted intensity variation by invoking modulation by charge screening and interference, at constant surface charge density, as detailed in ref. ²⁶.”

4) The auto-catalytic mechanism requires a net increase in catalyst (hydroxide) as the reaction progresses, but this is not implicitly shown as Figure 5 illustrates two hydroxides being used to generate one dissolved deprotonated silicic acid, which results in the generation of two hydroxides after protonation of the solution species (two used => two generated). I think the authors need to show how the remaining surface siloxide formed after release of the silicic acid can react with water

to form some kind of species like that formed in equation 3. This bypasses the need for hydroxide in that step which means that there would be a net increase in hydroxide once dissolution of that group takes place (one hydroxide used, two generated by the released silicic acid). In other words, the scheme shows hydroxide being catalytic, but another step needs to be added to show a net increase in hydroxide which backs up the observation of autocatalysis.

In agreement with the reviewer's comment, we have revised Figure 5 in the manuscript, which is reproduced below. The protonation / deprotonation equilibrium of the surface is far on the deprotonation side. Thus, most of the generated hydroxides (through dissolution) will be used to dissolve another deprotonated surface siloxide. As is clear from Fig. 5, by every consumption of OH⁻ for the dissolution, two OH⁻ are released. Subsequently, these two OH⁻ can dissolve two surface siloxide resulting in four OH⁻. We hope that the figure now demonstrate the autocatalytic process clearer.

Action: we have revised figure 5.

5) Page 2, line 37: should be "insights are scarce".

Action: In line with the reviewers suggestion we have changed "insights scarce" to "insights are scarce" in line 36.

6) The cited Reference 17 is only the correction to the paper and not the original paper. Perhaps both should be cited?

Action: According to the reviewer's comment, we have included the original publication in addition to the correction.

Response to reviewer 2

We thank the reviewer for careful reading the manuscript, and for the important and useful feedback that has helped improved the paper. Below, we give a point-by-point response to the comments of the reviewer, and the action taken. We have reproduced the reviewer's comments in italic font below.

The manuscript by Schaefer et al reports an investigation on dissociation of silica surface in contact with water by surface-specific vibrational spectroscopy. The authors evaluated the interfacial ionic concentration arisen from dissolution products and the associated reaction kinetics through examining time-dependent SFG intensity and the effect of liquid flow. While the experimental results appear to have far-reaching implications for interfacial chemistry, I have concerns about analysis and interpretation, as describing in details below. Since it is crucial for validity of the main conclusions of the manuscript, I think that the manuscript is not suitable for publication with the current form.

We appreciate the reviewer's assessment that the subject matter of our paper is important, and have performed additional experiments and analyses that corroborate the interpretation of our work. The reviewer raises legitimate concerns that we have tried to address in the revised manuscript.

- 1. The experiments in the manuscript are mainly explained by assuming an insignificant change in the surface charge density of the silica/water interface, but this assumption is not justified nor discussed in this manuscript. (nor in Ref. 24, where Fig. 1 of this manuscript comes from). In particular, the authors make this assumption in line 60 and 61 by considering the local (surface) pH as the primary factor determining the surface charge density. This argument is not sufficient to support the authors' assumption. It is because the surface pH is associated with the interfacial potential (Ref. 25), and the interfacial potential changes upon different ionic concentrations in the presented experiments. It is, therefore, questionable whether the variations of local pH and the consequent surface charge density can be assumed insignificant in interpreting the experiments. Note that possible changes of the surface charge density could affect the interfacial ionic concentration deduced in Fig. 4 (deduced under assumption of constant surface charge density) and could, therefore, affect validity of the authors' subsequent discussions about kinetics and pathway of the reaction (Fig. 5).*

We agree with the reviewer that the surface charge density is *a priori* not independent of the interfacial salt concentration. However, we have demonstrated in ref. 26 (Schaefer et al, PCCP, 2017) that the dependence of the SFG response on the ion concentration can be well explained without invoking substantial changes in the surface charge density. To illustrate this point, please consider the blue trace in the figure below, which is a no-adjustable-parameter description of the SFG intensity as a function of salt concentration, reproduced from our Ref. 26, for the same surface, assuming constant surface charge as a function of salt concentration.

Reprinted with permission from:
 J. Schaefer, G. Gonella, M. Bonn and E. H.
 G. Backus, Phys. Chem. Chem. Phys., 2017,
 19, 16875. DOI: 10.1039/C7CP02251DS
 Published by the PCCP Owner Societies.

This agreement between experimentally observed SFG intensity and constant-surface-charge theory, indicates that, while we cannot exclude a variation in surface charge, the effects on the SFG intensity due to screening dominate our experiments.

Action: We have added a statement along these lines on lines 59 – 61 in the revised manuscript: “In ref ²⁶, it is shown that the SFG intensity variation with ionic strength of NaCl can be accounted for by invoking only screening and assuming the surface charge, primarily determined by the local pH,²⁷ to be constant.²⁶”, and in the figure caption of figure 1: “Dashed line: The predicted intensity variation by invoking modulation by charge screening and interference, at constant surface charge density, as detailed in ref. ²⁶.”

Another indication that screening is the major contribution to the change in SFG response, following dissolution in pure water, comes from the frequency-resolved spectroscopy: we observe a spectral shift in the center-of-mass of the OH stretch band of $\sim 40 \text{ cm}^{-1}$ between pure water and a 1 mM salt solution in Schaefer et al (PCCP, 2017). We observe precisely this shift in the dissolution experiment of this work (shown below), for which we conclude, from the intensity, that a roughly $\sim 1 \text{ mM}$ solution of dissolution salts is present at the surface. Again, the spectral data are consistent with the conclusion from the intensity data that it is justified to assume a negligible change in surface charge.

Action: We have added this figure to the Supporting Information of the revised manuscript (S1) and mention the agreement between the intensity and spectral data in the revised manuscript in line 95-105, which also addresses point 3 of the reviewer “A spectral analysis of the signals supports the conclusion that the dissolution products in this concentration range screen the surface charge. The spectral shift of the O-H band corresponds quantitatively to that observed previously between pure water and a salt solution (NaCl) in the millimolar concentration range (detailed in Figure S1).²⁶ As mentioned above, the associated SFG intensity variation of those NaCl solutions can be accounted for by just invoking a concentration dependent Debye screening length, keeping the surface charge density constant (Figure 1, dashed line).²⁶ This suggests that the intensity change, also along dissolution, primarily results from variation in charge screening. However, we cannot exclude an additional contribution from changes in the surface charge, as indicated in a previous work using a different, more surface-specific, experimental geometry.²⁵”

We note, however, that a control experiment of dissolution at pH 9– a new experiment that we have added in the Supporting Information of the revised manuscript, following the request of reviewer 3 (shown in the new Figure S4) — shows that a change of surface charge is likely relevant for that case. In other words: the situation at pH = 7 is not general, and in that sense the reviewer is completely right in requesting a better justification of the assumption of constant surface charge. Also, different experimental geometries are differently sensitive to changes in the surface charge relative to changes in the screening (see also our reply to point 3 below).

2. *The auto-catalytic process proposed in the manuscript generates a negatively charged Si-O-group and an over-protonated monosilicic acid during one cycle of the reaction (Eq. 3-5). The authors used screening effect of the over-protonated monosilicic acid to explain the time-dependent SFG measurement in Fig. 2. The negatively charged byproducts, i.e., negatively charged Si-O- groups, with the equal amount are then expected to appear at the interface and thus contribute to the surface charge density. It is unclear whether or not the surface Si-O- groups generated from the auto-catalytic process induce minor changes of the surface charge density, as assumed in the manuscript. This issue needs to be addressed to ensure self-consistency of the data interpretation.*

The reviewer raises a good point. We have revised Figure 5, to stress that there is no net change of surface charge along dissolution. The revised figure 5 is reproduced here:

Since the protonation / deprotonation equilibrium of surface silanols is far on the deprotonation side, it is largely unaffected by the generation of OH^- which is supported by previous SFG studies of this interface (e.g. DeWalt-Kerian et al, JPCL, 8, 2017) showing that the SFG response is relatively insensitive to pH at pH=7. In contrast, the dissolution step is far on the educt side and may “consume” OH^- directly when generated. The net result is that no charge is generated, as shown in the schematic below; the difference between initial state and equilibrium is the localization of dissolved positive charge: Initial state: Proton (fast diffusion) vs. Equilibrium: Silicic acid (slow diffusion; stays longer at the interface, thereby screening the negative surface charge).

Action: we have revised figure 5.

3.

(a) The authors have to address (in)consistency of results presented in the manuscript with respect to these in their previous reports. Some important information are missing. For instance, a figure from Ref. 24 is used as Fig. 1 in the presented manuscript to discuss the salt-concentration effect. It is unclear whether Fig. 1 was measured under resting or flowing condition, and whether Fig. 1 and Fig. 2 are comparable with each other.

We apologize for not providing all information in the manuscript. Fig. 1 is measured under resting condition directly after flowing for 5 minutes, as described in the experimental part. We now mention it also in the caption of Fig. 1.: "All spectra are measured under resting conditions, directly after flowing for 5 minutes." Moreover, Fig. 1 and 2 can be compared, the spectra

presented in Fig. 1 correspond to the initial state that is the first spectrum after turning off the flow in Fig. 2.

(b) In addition, in a previous paper published by the same group (ref 23), the authors observed ~50% difference in the SFG intensity under resting and flowing conditions in similar experimental conditions (silica/water interface, 10mM NaCl, pH=6.5, flow OFF and ON). However, this effect is obviously weaker in the submitted manuscript. (Fig. 2 in yellow, flow OFF and ON at 10 mM, and Fig. 4 at 10mM, the difference is only ~10%). It is also noticed that the data interpretations are different in the authors' different publications. In Ref. 23, the difference of the SFG intensity observed upon flowing of liquids was explained by the change in the surface charge density, and the change of Debye length was considered to be minor. These interpretations are obviously against these in the presented manuscript, without any comment given.

We are very grateful for this important point, which also connects to the first concern of the reviewer, regarding the role of variations in surface charge on the signals. The main difference between the present paper and our previous paper (ref. 25) that the reviewer refers to, is the experimental geometry: In the present experiments, we employ a thin silica plate in reflection geometry, whereby the bulk of the IR light is transmitted through the sample plate. In contrast, in reference 23, we employed a prism geometry, which uses an evanescent wave to probe the surface. In that case, also the IR field is strongly z-dependent (in addition to the surface electric field) which changes the ion concentration dependence of the SFG signal and makes it overall more surface specific. In the literature, this difference is apparent by comparing Schaefer et al, PCCP, 2017 vs Jena et al, JPCL 2011: in the former, which uses the same geometry as in the present paper, the SFG intensity is seen to drop off abruptly for salt concentrations below ~1 mM:

Reprinted with permission from:
 J. Schaefer, G. Gonella, M. Bonn and E. H. G. Backus, Phys. Chem. Chem. Phys., 2017, 19, 16875. DOI: 10.1039/C7CP02251DS
 Published by the PCCP Owner Societies.

This was not observed in the latter publication, which also uses a prism geometry (figure 3 from Jena et al., JPCL 2011). In addition, the interfacial layer (~region C) contributes with ~40% to the maximum signal while it is only about 10% in the former publication:

Reprinted with permission from K.C. Jena, P.A. Covert, and D. K. Hore*
 J. Phys. Chem. Lett., 2011, 2 (9), pp 1056–1061. DOI: 10.1021/jz200251h.
 Copyright 2011 American Chemical Society

Action: We have added a statement regarding the assumption of constant charge on lines 59 – 61 in the revised manuscript: “In ref ²⁶, it is shown that the SFG intensity variation with ionic strength of NaCl can be accounted for by invoking only screening and assuming the surface charge, primarily determined by the local pH,²⁷ to be constant.²⁶”, and in the figure caption of figure 1: “Dashed line: The predicted intensity variation by invoking modulation by charge screening and interference, at constant surface charge density, as detailed in ref. ²⁶.”

Action: A discussion justifying the assumption of constant charge, and a comparison to our previous work has been added to lines 95-105: “A spectral analysis of the signals supports the conclusion that the dissolution products in this concentration range screen the surface charge. The spectral shift of the O-H band corresponds quantitatively to that observed previously between pure water and a salt solution (NaCl) in the millimolar concentration range (detailed in Figure S1).²⁶ As mentioned above, the associated SFG intensity variation of those NaCl solutions can be accounted for by just invoking a concentration dependent Debye screening length, keeping the surface charge density constant (Figure 1, dashed line).²⁶ This suggests that the intensity change, also along dissolution, primarily results from variation in charge screening. However, we cannot exclude an additional contribution from changes in the surface charge, as indicated in a previous work using a different, more surface-specific, experimental geometry.²⁵”

Response to reviewer 3

We thank the reviewer for careful reading the manuscript, and for providing feedback that has helped improve the manuscript. Below, we give a point-by-point response to the comments of the reviewer and the action taken. We have reproduced the reviewer's comments in italic font below.

This paper presents some very interesting experiments in which vibrational sum-frequency generation is used to probe the silica/water interface over a time scale of many hours as the silica is slowly dissolved. The authors present convincing kinetic data showing that dissolution is considerably faster than has been concluded in other studies, which have focused on bulk rather than interfacial observables. This is a clever and fascinating study that will undoubtedly receive a lot of well-deserved attention.

We are grateful for the reviewer's positive assessment of our study.

The part of the work that I am less convinced about is the argument for autocatalysis. The authors observe that the rate of dissolution increases after an induction period, which they convincingly argue cannot be described by the simplest reasonable kinetic model. They propose a plausible mechanism to account for this observation. However, if autocatalysis is going to be a primary focus of the paper (and it is in the title, after all), then some control experiments are called for.

First, as they posit that base-catalyzed hydrolysis accounts for their observations, it would seem that if they were to begin their experiments under basic conditions that there would be no induction period.

In accordance with the reviewer's suggestion, we performed a control experiment with a pH 9 aqueous solution of NaOH which is now mentioned in lines 206-210 and included in the Supporting Information (Figure S4). The figure is reproduced here. In line with our autocatalytic picture, we observe that the kinetics speed up by over an order of magnitude: from tens of hours to a 1-hour timescale under basic conditions. This qualitative conclusion supports our interpretation, and we are grateful for the reviewer for suggesting this experiment.

We have chosen not to include this measurement in the main text, since we are unable to quantitatively interpret this experiment without a full calibration measurement of the SFG signal intensity versus salt concentration at this particular pH. This is necessary since an increase of pH

shifts all the involved equilibria (silanol deprotonation, dissolution and protonation ratios of dissolved species).

Action: We have added this new measurement as figure S4 to the Supporting Information, including a discussion.

Second, I am not sure what the plasma treatment does to the surface of the silica, but one possibility is that it tends to remove all of the silanol functionality (although this may be restored by contact with water). At any rate, the induction period could certainly have something to do with the surface structure of the silica following plasma treatment, and it is not at all clear to me that the rate-limiting step should be the final one in dissolution. Each Si atom in the bulk is bonded to four O atoms, and I would (admittedly, possibly naively) think that the breaking of the first siloxane bridge for a particular Si atom would be rate limiting. I would suggest two control experiments to test the proposed mechanism further.

The first of these the authors may already have performed, which is to determine what happens if they allow the dissolution to reach its maximum rate and then flow pure water into the system. Are the kinetics then reproducible, including the induction period? The second control would be to use a high-temperature vacuum oven instead of plasma treatment to prepare the surface. The vacuum oven will also remove organic contaminants, but will also tend to remove silanol groups so that all of the surface oxygen atoms are involved in siloxane bridges. Looking at the effects of this alternate treatment should reveal if the induction period is related to surface structure.

In agreement with the reviewer's comment, we performed a control experiment, where we performed several consecutive cycles of flow-off / flow-on. As presented in the figure below, the kinetics are well reproduced with each cycle: both the timescale (tens of hours) and the induction period (several hours) are reproduced in the different measurements. The fact that we can reproduce the measurements over a time scale of several days on one and the same surface, indicates that the kinetics are largely unaffected by the history of the surface, i.e. the cleaning procedure. Over such long time scales, some surface contamination cannot be avoided, which likely explains the overall decrease in signal strength with time. An alternative cleaning procedure, using a high-temperature vacuum oven, is more problematic since it is still debated in the community if this treatment de- or increases the silanol population at the silica surface. (see Schrade et al., PNAS, 2018 vs. Dalstein et al., PCCP, 2017).

On a minor additional note, the authors should do another careful proofreading of the paper, as I did come across a number of typographical errors.

Action: In accordance with the reviewer's suggestion, we did another careful proofreading of the manuscript, and have made several adjustments.

REVIEWERS' COMMENTS:

Reviewer #1 (Remarks to the Author):

The authors have satisfied my concerns, and the manuscript is now suitable for publication.

Reviewer #2 (Remarks to the Author):

The authors have made careful revision of the manuscript and provided detailed explanations in response to the referee's comments. I still have a concern about the first comment in the original referee report, as described in details below. If further revision can be made, I will recommend publication in Nature Communication.

Comment 1:

Agreement between the salt-concentration-dependent SFG intensity and the theoretical fit in Fig. 1 supports validity of constant surface charge density assumed. This agreement is also crucial as being the basis for attributing the time-dependent intensity change to changes in the ion concentration. The authors have to address all assumptions in this theoretical fit explicitly for avoiding possible misuse of the presented results by readers. In particular, the authors assume constant $\chi(2)$ in this theory, while $\chi(2)$ is expected to change in response to dissociation of surface silanol groups. The authors have to indicate the assumption of $\chi(2)$, together with the assumption of the constant surface charge density, throughout the paper (e.g., line 61, 72, and 102).

Reviewer #3 (Remarks to the Author):

The authors have done an excellent job of responding to my comments and concerns, as well as those of the other reviewers. This is an important piece of work that is destined to be highly cited, and I recommend publication in its current form.

Reviewer #1 (Remarks to the Author):

The authors have satisfied my concerns, and the manuscript is now suitable for publication.

Response to reviewer 1:

We thank the reviewer for carefully reading of the revised manuscript and the supporting comment.

Reviewer #2 (Remarks to the Author):

The authors have made careful revision of the manuscript and provided detailed explanations in response to the referee's comments. I still have a concern about the first comment in the original referee report, as described in details below. If further revision can be made, I will recommend publication in Nature Communication.

Response to reviewer 2:

We thank the reviewer for carefully reading of the revised manuscript and the constructive comment. We have reproduced the reviewer's comment in italic font below.

Comment 1:

Agreement between the salt-concentration-dependent SFG intensity and the theoretical fit in Fig. 1 supports validity of constant surface charge density assumed. This agreement is also crucial as being the basis for attributing the time-dependent intensity change to changes in the ion concentration. The authors have to address all assumptions in this theoretical fit explicitly for avoiding possible misuse of the presented results by readers. In particular, the authors assume constant $\chi(2)$ in this theory, while $\chi(2)$ is expected to change in response to dissociation of surface silanol groups. The authors have to indicate the assumption of $\chi(2)$, together with the assumption of the constant surface charge density, throughout the paper (e.g., line 61, 72, and 102).

The reviewer raises a good point. According to the reviewer's comment, we have added the assumption of constant $\chi(2)$ in addition to the constant surface charge density in the mentioned lines.

Reviewer #3 (Remarks to the Author):

The authors have done an excellent job of responding to my comments and concerns, as well as those of the other reviewers. This is an important piece of work that is destined to be highly cited, and I recommend publication in its current form.

Response to reviewer 3:

We thank the reviewer for carefully reading of the revised manuscript and the supporting comment.